# Screening, Characterization and Comparison of Endoglucanases/Xylanases from Thermophilic Fungi: A *Thielavia terrestris* Xylanase with High Activity-Stability Properties

**DOI:** 10.3390/ijms26146849

**Published:** 2025-07-17

**Authors:** Shaohua Xu, Kexuan Ma, Zixiang Chen, Jian Zhao, Xin Song, Yuqi Qin

**Affiliations:** 1National Glycoengineering Research Center, Shandong University, Qingdao 266237, China; sh15064159636@163.com (S.X.); makexuan@mail.sdu.edu.cn (K.M.); songx@sdu.edu.cn (X.S.); 2State Key Laboratory of Microbial Technology, Shandong University, Qingdao 266237, China; 202212562@mail.sdu.edu.cn (Z.C.); zhaojian@sdu.edu.cn (J.Z.)

**Keywords:** cellulases, fungi, glycoside hydrolases, hemicellulases, xylanases, thermophilic fungi

## Abstract

Thermostable cellulases and xylanases have broad acceptance in food, feed, paper and pulp, and bioconversion of lignocellulosics. Thermophilic fungi serve as an excellent source of thermostable enzymes. This study characterized four endo-β-1,4-glucanases (two glycoside hydrolase (GH) family 5 and two GH7 members) and four endo-β-1,4-xylanases (two GH10 and two GH11 members) from thermophilic fungus *Thielavia terrestris*, along with one GH10 endo-β-1,4-xylanase each from thermophilic fungus *Chaetomium thermophilum* and mesophilic fungus *Chaetomium globosum*. Comparative analysis was conducted against three previously reported GH10 endoxylanases: two thermostable enzymes from the thermophilic fungus *Humicola insolens* and thermophilic bacterium *Halalkalibacterium halodurans*, and one mesophilic enzyme from model fungus *Neurospora crassa*. The GH10 xylanase TtXyn10C (Thite_2118148; UniProt G2R8T7) from *T. terrestris* demonstrated high thermostability and activity, with an optimal temperature of 80–85 °C. It retained over 60% of its activity after 2 h at 70 °C, maintained approximately 30% activity after 15 min at 80 °C, and showed nearly complete stability following 1 min of exposure to 95 °C. TtXyn10C exhibited specific activity toward beechwood xylan (1130 ± 15 U/mg) that exceeded xylanases from *H. insolens* and *H. halodurans* while being comparable to *N. crassa* xylanase activity. Furthermore, TtXyn10C maintained stability across a pH range of 3–9 and resisted trypsin digestion, indicating its broad applicability. The study expands understanding of enzymes from thermophilic fungi. The discovery of the TtXyn10C offers a new model for investigating the high activity-stability trade-off and structure-activity relationships critical for industrial enzymes.

## 1. Introduction

Lignocellulosic biomass, Earth’s most abundant renewable resource, consists primarily of the macromolecular polymers cellulose, hemicellulose, and lignin. Cellulose, a linear β-1,4-glucan polymer composed of glucose units, forms intricate structures featuring highly ordered crystalline and amorphous regions [1]. Its degradation is initiated by endo-β-1,4-glucanases (EC 3.2.1.4), which hydrolyze internal β-1,4-glycosidic bonds within the cellulose chains. Xylan, the predominant structural polymer in hemicellulose, possesses a backbone of β-1,4-linked D-xylopyranose residues [2]. Cleavage of internal β-1,4-glycosidic bonds within the xylan backbone is catalyzed by endo-β-1,4-xylanases (EC 3.2.1.8), recognized as the most critical enzymes in hemicellulose-degrading systems [3].

Currently reported endo-β-1,4-glucanases are distributed across 14 glycoside hydrolase (GH) families: GH5, 6, 7, 8, 9, 12, 26, 44, 45, 48, 51, 74, 124, and 148 (http://www.cazy.org/ (accessed on 10 April 2025)). Among these, GH5 and GH7 family endoglucanases are particularly significant for textile, pulp and paper, laundry, and industrial biomass conversion [4,5]. For instance, in the renowned cellulase-producing fungus *Trichoderma reesei*, its two most abundant endoglucanases—Cel7B (formerly EGI) from GH7 and Cel5A (formerly EGII) from GH5—synergistically cleave the internal cellulose chain to reduce substrate polymerization degree and generate new chain ends for cellobiohydrolases (CBHs) [6,7].

Currently reported endo-β-1,4-xylanases are distributed across 14 glycoside hydrolase (GH) families: GH3, 5, 6, 8, 9, 10, 11, 30, 43, 44, 51, 62, 98, and 141 (http://www.cazy.org/ (accessed on 10 April 2025)). Particularly, GH10 and GH11 family xylanases exhibit broad industrial applications in the food, feed, pulp and paper, biofuels, and textile sectors [8,9]. These enzymes exhibit distinct molecular weight, structure, substrate specificities, and catalytic mechanisms [10]. GH10 xylanases typically exhibit molecular weights >40 kDa, while GH11 enzymes generally range around 20 kDa and rarely >30 kDa. GH10 xylanases have a characteristic (α/β)_8_ TIM-barrel fold formed by eight parallel β-strands surrounded by eight α-helices [11,12]. They typically possess wide and shallow active-site clefts, conferring broad substrate versatility. This topology permits extensive conformational flexibility, allowing the accommodation of highly substituted xylans. GH11 xylanases have a characteristic β-jelly-roll fold formed by two antiparallel β-sheets and one α-helix [13,14]. They possess elongated and narrow active-site clefts that restrict access to glycosidic bonds near branch points, preferentially targeting unsubstituted regions of the xylan backbone [15].

While mesophilic enzymes from mesophilic fungi like *Trichoderma* and *Aspergillus* dominate industrial applications, thermostable cellulases/xylanases derived from thermophilic fungi offer significant advantages, including enhanced reaction rates at elevated temperatures, improved stability, and longer shelf lives [16,17,18]. Thermostable xylanases can be utilized in the food industry to enhance dough properties and bread quality [19] and in the feed industry to improve animal health and productivity [20]. Thermophilic fungi, such as *Chaetomium thermophilum*, *Humicola insolens*, *Myceliophthora thermophila*, and *Thielavia terrestris*, can thrive at high temperatures (45–60 °C) and produce various thermostable enzymes [16,21,22]. For example, *C. thermophilum*, as the first thermophilic fungal genome sequenced [23], produces diverse CAZymes, including thermostable GH7/GH45 endoglucanases and GH11 endoxylanases [24,25,26].

*T. terrestris* (syn. *Thermothielavioides terrestris*) encodes over 210 glycoside hydrolases (GHs) and secretes over 180 carbohydrate-active enzymes (CAZymes). Supplementation with *T. terrestris* extracellular enzymes enhanced the saccharification of mild steam-pretreated spruce when combined with a commercial enzyme cocktail [27]. Many extracellular GHs demonstrated thermostability—such as a GH93 exo-arabinanase showing peak activity at 70 °C that synergizes with arabinofuranosidase for arabinose/arabinosaccharide production from beet pulp [28], a feruloyl esterase maintaining 84.16% activity after 1 h at 60 °C that cooperates with xylanase to release ferulic acid from wheat bran [29], and a GH45 endocellulase retaining >80% activity after 2.5 h at 80 °C [30]. These results underscore *T. terrestris* as an excellent source of thermostable enzymes. However, despite this significant potential, the functional characterization of endo-β-1,4-glucanases/xylanases in *T. terrestris* lacks detailed biochemical profiling.

This study characterized thirteen endo-β-1,4-glucanases/xylanases, including four endoglucanases (two GH5 and two GH7 members) and four endoxylanases (two GH10 and two GH11 members) from *T. terrestris*, along with five other GH10 endoxylanases from thermophilic/mesophilic fungi or bacteria. Comparative analysis was performed among GH10 endoxylanases. Notably, the GH10 xylanase TtXyn10C (Thite_2118148; UniProt Entry: G2R8T7) from *T. terrestris* demonstrated exceptional thermostability and activity.

## 2. Results

### 2.1. The Selection of Target Endo-β-1,4-Endoglucanases/Xylanases in T. terrestris

The genome of *T. terrestris* NRRL 8126 harbors 11 endo-β-1,4-endoglucanase-encoding genes, classified into the GH5, GH7, GH12, and GH45 families. We named five GH5 endoglucanases as TtCel5A-E, three GH7 endoglucanases as TtCel7A-C, one GH12 endoglucanase as TtCel12A, and two GH45 endoglucanases as TtCel45A/B (Figure 1A, yellow background). There are 11 endo-β-1,4-xylanase-encoding genes, classified into the GH10 and GH11 families. We named six GH10 endoxylanases as TtXyn10A-F and five GH11 endoxylanases as TtXyn11A-E (Figure 1A, blue background). TtCel5C, TtCel7A/C, and TtXyn10A/C were detectable in the secretome of *T. terrestris* when cultured with either alfalfa or barley straw as the carbon source [16]. The GH family classification, protein IDs, and designated nomenclature for all 22 endoglucanases/xylanases are listed in Figure 1A.

Transcriptomic analysis revealed differential expression among these endo-β-1,4-endoglucanases/xylanases genes, with TtCel5A/5C, TtCel7A/7C, TtXyn10A/10B/10C, and TtXyn11C/11D/11E displaying significantly higher transcript levels than the other genes when *T. terrestris* was grown on alfalfa or barley straw as the carbon source (Figure 1B). Given their high expression levels or secretion, we selected four endoglucanases (TtCel5A, TtCel5C, TtCel7A, and TtCel7C) and four endoxylanases (TtXyn10B, TtXyn10C, TtXyn11C, and TtXyn11D) for further investigation.

### 2.2. Recombinant TtCel5A and TtXyn10B/10C Exhibit High Optimum Temperatures

After expressing eight selected endo-β-1,4-glucanases/xylanases in *Pichia pastoris*, their optimal temperature and pH were investigated. For the four endo-β-1,4-glucanases, pH optimum assays revealed optimal activity at pH 5.0 for TtCel5A and TtCel5C and at pH 6.0 for TtCel7A and TtCel7C (Figure 2A). Temperature profiling revealed that only TtCel5A exhibited an optimum temperature of 70 °C, while the optimum temperatures of the other three endoglucanases were all below 70 °C (Figure 2B). However, TtCel5A displayed unexpectedly low thermostability, retaining only approximately 20% residual activity after 30 min of incubation at 60 °C (Figure 2C).

For the four endo-β-1,4-xylanases, optimal activity occurred at pH 5.0 for TtXyn10B and TtXyn10C and at pH 4.0 for TtXyn11C and TtXyn11D; notably, while TtXyn11C and TtXyn11D exhibited significantly reduced activity at pH 6.0, TtXyn10B and TtXyn10C demonstrated broader pH adaptability, retaining over 80% activity at pH 6.0 and >50% at pH 7.0 (Figure 2D). Temperature profiling indicated optimal activities at 60 °C to 70 °C for TtXyn10B, 80 °C to 85 °C for TtXyn10C (retaining >90% activity at 85 °C), and 60 °C for both TtXyn11C and TtXyn11D (Figure 2E).

### 2.3. Recombinant TtXyn10C Exhibits Outstanding Thermostability Properties

Given their higher optimal temperatures, the recombinant TtXyn10B and TtXyn10C were further characterized. SDS-PAGE analysis revealed that recombinant TtXyn10B exhibited a single band corresponding to its theoretical molecular weight of 40.4 kDa, whereas recombinant TtXyn10C migrated as a broad smear between 45 and 70 kDa (Figure 3A), indicating molecular heterogeneity. After Endo H treatment, the diffuse band of TtXyn10C resolved into a single band (Figure 3B), demonstrating that this heterogeneity was attributable to glycosylation of the recombinant protein. The specific activity of deglycosylated TtXyn10C (dTtXyn10C) decreased by approximately 24.6% compared to the glycosylated TtXyn10C. Kinetic analysis of TtXyn10C activity toward beechwood xylan yielded a *K*_m_ value of 0.48 mg/mL, a *V*_max_ value of 10,561 nkcat/mg, and a *K*_cat_ value of 21.12 s^−1^. In-gel activity staining using separating gels containing beechwood xylan revealed clear zones unstained by Congo red, demonstrating their efficient degradation of beechwood xylan. TtXyn10B and TtXyn10C exhibited electrophoretic band smearing with extended hydrolytic zones beyond their predicted molecular weights, likely due to enzyme adsorption to the xylan substrate during electrophoretic migration (Figure 3C).

TtXyn10B demonstrated moderate thermostability, retaining approximately 80% of initial activity after 8 h incubation at 60 °C. However, its thermal tolerance decreased sharply at higher temperatures, with complete inactivation within 15 min at 70 °C and within 5 min at 80 °C (Figure 3D). TtXyn10C exhibited exceptional thermal stability. The enzyme maintained nearly 100% of its original activity following incubations of 8 h at 37 °C, 50 °C, and 60 °C. Even under more extreme conditions, TtXyn10C showed remarkable thermostability, retaining over 60% activity after 2 h at 70 °C and maintaining approximately 30% activity after 15 min at 80 °C. EndoH treatment resulted in a slight decrease in the stability of TtXyn10C; the thermostability of dTtXyn10C retains about 50% activity after 2 h at 70 °C and maintains approximately 10% activity after 15 min at 80 °C (Figure 3E).

### 2.4. Recombinant TtXyn10C Shows High Enzymatic Activity

Five representative GH10 xylanases from distinct organisms were selected for comparative analysis with TtXyn10C, including (1) XynA (AGG68962.1) from *H. insolens* Y1—a thermophilic model fungus. The *H. insolens* XynA has remarkable thermostability, broad substrate specificity, and alkaline tolerance (pH 8–10) with high catalytic efficiency [31]. (2) XynA (AP07528) from *H. halodurans* (Syn. *Bacillus halodurans*)—a hyperthermophilic bacterium. *H. halodurans* XynA is a thermostable (70–75 °C) and alkali-tolerant (pH 5–9) xylanase [32]. (3) NCU08189 (XP_959279.1) from *Neurospora crassa*—a model mesophilic fungus. The optimal temperature of NCU08189 is 40–65 °C [33]. (4) A xylanase (XP_006694397.1) from *C. thermophilum*. The thermophilic fungus *C. thermophilum* produces thermostable GH7 endoglucanases and GH11 endoxylanases [24,26]. However, its GH10 endoxylanases are not yet characterized. (5) A xylanase (KAH6635743.1) from mesophilic *C*. *globosum* serves as phylogenetic control for CtXyn10. In this study, the five xylanases above were designated HiXyn, HhXyn, NcXyn, CtXyn, and CgXyn, respectively. Their protein IDs, identities with TtXyn10C, predicted molecular weights, and references are listed in Figure 4A.

The recombinant HiXyn, NcXyn, HhXyn, CtXyn, and CgXyn were purified (Figure 4B). The specific activity of five xylanases was evaluated using beechwood xylan as a substrate (Figure 4C). The specific activities of TtXyn10C, HiXyn, NcXyn, HhXyn, CtXyn, and CgXyn are 1129.8 ± 15.3 U/mg, 354.4 ± 30.3 U/mg, 1153.2 ± 29.1 U/mg, 319.8 ± 3.8 U/mg, 362.6 ± 13.9 U/mg, and 507.9 ± 17.3 U/mg, respectively. Except for NcXyn10C, which exhibited slightly higher activity than TtXyn10C, all others showed lower activity than TtXyn10C. The bacterial-derived HhXyn displayed the lowest activity. Their optimal pHs and temperatures were assayed. The optimal pHs of CgXyn and CtXyn are pH6. The optimal pHs of HiXyn and NcXyn are also pH6, as reported, with HiXyn demonstrating unusual alkaline persistence (>50% activity at pH 7–9) [31,33]. The optimal pH of HhXyn is pH9.0, consistent with prior characterization [32,34] (Figure 4D). HiXyn and CtXyn displayed optimal activity at 70 °C. NcXyn, HhXyn, and CgXyn showed optimal temperature at 60 °C. All tested enzymes exhibited lower thermal optima than TtXyn10C (Figure 4E).

### 2.5. Recombinant TtXyn10C Exhibits Stability Across a Broad pH Range, Brief High-Temperature Exposure, and Trypsin

In practical applications, the animal feed sector constitutes the largest market share (42%) for xylanases [35]. To evaluate the application potential of TtXyn10C in feed, its pH stability, tolerance to brief high-temperature exposure, and resistance to trypsin were assessed.

The stability of TtXyn10C was assessed across pH 2.0 to 9.0. The enzyme was unstable at pH 2.0 but demonstrated remarkable stability between pH 3.0 and 9.0. Activity remained largely unchanged (>95%) after 2 h incubation at pH 4.0–8.0. At pH 9.0 and pH 3.0, it retained 86% and 93% of its initial activity after 2 h incubation, respectively (Figure 5A). TtXyn10C exhibited significant resistance to short-term thermal stress, retaining nearly full activity after 1 min incubation at temperatures ranging from 75 °C to 95 °C (Figure 5B). The enzyme also demonstrated strong resistance to trypsin digestion (Figure 5C).

### 2.6. Structural Comparison of TtXyn10C and CtXyn Proteins

TtXyn10C exhibited high thermostability, but its homolog in *C. thermophilum*, CtXyn, showed limited heat resistance despite also originating from a thermophilic organism: CtXyn retained only 16% of its initial activity after 15 min at 70 °C and was inactivated entirely after 120 min (Appendix A). This difference is interesting, as there is high identity (77%) between TtXyn10C (351 residues) and CtXyn (350 residues). Given the absence of experimentally determined structures for both TtXyn10C and CtXyn, we used homology modeling to compare their protein architectures and performed an analysis of putative thermostability and activity determinants.

Structural modeling confirmed both GH10 xylanases adopt the canonical (β/α)_8_ TIM-barrel fold, featuring eight parallel β-strands forming the central barrel surrounded by eight α-helices (Figure 6A). Molecular docking with xyloseptaose substrate revealed near-identical substrate-binding environments. Most essential residues within 5 Å of the ligand were strictly conserved between TtXyn10C and CtXyn, including the catalytic Glu166 (general acid/base) and Glu273 (nucleophile), along with Tyr209 interacting with the subsite +1 substrate and the Trp121-Trp313-Trp321 aromatic triad at subsite −1 comprising the substrate-stacking aromatic cage (Figure 6B) [11].

Despite this high conservation, four structural variations were identified (Figure 6C). Notably, the TtXyn10C-Tyr213/CtXyn-Trp213 divergence is particularly significant, as tyrosine conservation at the equivalent position (Tyr272) in the thermostable homolog *Bispora* sp. XYL10C has been demonstrated to mediate essential π-stacking interactions with substrates [36]. Similarly, the TtXyn10C-Ser325/CtXyn-Tyr325 substitution suggests evolutionary optimization, with serine occupying the homologous position (Ser384) in XYL10C [36]. At position 248, the TtXyn10C-Ser248/CtXyn-Asn248 replacement may confer dual functional advantages: (1) Ser248 eliminates the deamidation risk associated with Asn at elevated temperatures, thereby enhancing thermal stability; (2) its smaller hydroxyl group reduces steric hindrance while preserving hydrogen-bonding capacity, potentially optimizing substrate positioning for improved hydrolytic efficiency. These key residues likely represent critical determinants of TtXyn10Cs enhanced thermostability and catalytic efficiency.

## 3. Discussion

Thermophilic fungi represent a promising source of thermostable enzymes; however, we observed that among the eight endo-β-1,4-glucanases/xylanases from *T. terrestris*, only TtXyn10C exhibited excellent thermostability. Including four reported characterized enzymes, just 3 of 12 enzymes demonstrate thermostability: GH45 TtCel45A retaining >80% activity after 80 °C/2.5 h [30], TtXyn10A with a 23.1 days half-life at 65 °C [37], and TtXyn10C from this work. These initial observations give rise to the speculation that thermostable enzymes may constitute about 20% of the *T. terrestris*-produced lignocellulose-degrading enzymes.

While *T. terrestris* yields thermostable enzymes including endoglucanase TtCel45A [30], GH7 cellulase/xylanase bifunctional enzyme [38], β-galactosidase [39], exo-arabinanase [28], feruloyl esterase [29], and arabinofuranosidase [40], multiple studies report non-thermostable enzymes such as GH7 cellobiohydrolase [41], GH15 glucoamylase [42], acidic cutinase [43], and six lytic polysaccharide monooxygenases [44]. Moreover, although TtXyn10C exhibited high thermostability, its homolog, CtXyn, showed limited heat resistance despite also originating from a thermophilic organism. Given that even minor amino acid substitutions can significantly alter enzyme thermostability [45], this underscores that discovering thermostable enzymes from thermophilic fungi still requires exhaustive screening and characterization.

Xylanases are widely used in animal feed to enhance nutrient bioavailability, reduce anti-nutritional factors, and improve livestock gut health [46]. Feed enzymes require stringent performance characteristics: excellent thermostability to withstand brief high-temperature exposure during feed pelleting; stability across a broad pH range to function effectively throughout the gastrointestinal tract; and strong resistance to endogenous proteases [47,48]. TtXyn10C demonstrates high activity, exceptional stability across broad pH ranges, brief high-temperature exposure, and resistance to trypsin digestion, suggesting its potential as a promising candidate for feed enzyme applications pending further research. Additionally, given that many studies have demonstrated synergistic effects between xylanases (GH10/GH11) and cellulases in complex biomass degradation [49,50,51], future studies should investigate whether TtXyn10C exhibits synergy with cellulases and auxiliary enzymes under industrially relevant conditions to evaluate its potential for broader applications.

Compared to classical thermophilic reference enzymes, such as the HiXyn10 from thermophilic fungus *H. insolens* and HhXyn from thermophilic bacterium *H. halodurans*, TtXyn10C demonstrates not only exceptional thermostability but also superior catalytic activity. This dual superiority represents valuable property in thermophilic enzymes, where increased activity typically correlates with decreased stability. Although certain enzymes, such as XYL10C from *Bispora*, exhibit both exceptional thermostability and catalytic activity [52], the ‘activity-stability trade-off’ remains a pervasive phenomenon [53,54]. For instance, among characterized GH10 xylanases, sco5931 from *Streptomyces coelicolor* displays the highest reported specific activity (28,944 U/mg) yet shows limited thermal tolerance—retaining <50% residual activity after just 30 min at 60 °C [55].

The stability-activity trade-off originates from fundamental structural constraints: catalytic residues inherently act as stability weaknesses, while surrounding residue shells provide compensatory strengths through oscillatory free-energy patterns [56]. This necessitates a precise balance between global rigidity for stability and localized flexibility for catalytic function, particularly in thermophilic enzymes, where reduced scaffold flexibility enhances thermostability [45]. Simultaneous optimization remains challenging, as activity-enhancing mutations often destabilize proteins [54], while backbone rigidification can impede substrate-binding dynamics [57]. For example, rigidifying xylanase’s “cord” region improves stability but requires preserved “thumb” mobility to mitigate activity loss [58]. Optimizing the dynamics of the catalytic region and the rigidity of the noncatalytic region can simultaneously increase activity and stability in GH11 xylanase [59]. Recent researchers developed an isothermal compressibility-assisted dynamic squeezing index perturbation engineering (iCASE) strategy, which further enables synergistic gains. This strategy establishes region-specific optimization as a promising framework for robust biocatalyst design that overcomes the stability-activity trade-off [60].

Simultaneous attainment of high activity and stability represents a critical goal for industrial biocatalysis. Understanding the structure-dynamics-activity relationship is paramount for optimizing enzymatic performance under industrial constraints [61]. Future studies should elucidate the structural mechanisms conferring dual high-activity/thermostability traits in enzymes like *Bispora* sp. XYL10C and *T. terrestris* TtXyn10C, thereby informing rational design strategies.

## 4. Materials and Methods

### 4.1. Strains and Plasmids Construction

The encoding sequences of four *T. terrestris* endoglucanases (TtCel5A, TtCel5C, TtCel7A, TtCel7C) and four endoxylanases (TtXyn10B, TtXyn10C, TtXyn11C, TtXyn11D), along with five xylanases (HiXyn, HhXyn, NcXyn, CtXyn, CgXyn) from *H. insolens*, *H. halodurans*, *N. crassa*, *C. thermophilum*, and *C. globosum,* were synthesized by GenScript Biotech Corporation (Nanjing, China) with codon optimization for heterologous expression in *P. pastoris*. All sequences featured a C-terminal 6×His tag. Corresponding UniProt entries and GenBank accessions of the above enzymes are provided in Figure 1A and Figure 4A. These sequences were cloned into the *Sna*BI and *Eco*RI sites of plasmid pPIC9K. The recombinant plasmid was linearized with *Sac*I and transformed into *P. pastoris* GS115 cells.

### 4.2. Protein Expression and Purification

Recombinant *P. pastoris* strains expressing endoendoglucanases/xylanases were cultured according to the *Pichia* Expression Manual (Invitrogen, Carlsbad, CA, USA). Briefly, a single colony of *P. pastoris* was used to inoculate 25 mL of BMGY medium in a 250 mL baffled flask. The culture was grown at 28–30 °C with shaking at 300 rpm until reaching OD_600_ = 2–6 (approximately 16–18 h). Cells were harvested by centrifugation at 3000× *g* for 5 min at room temperature. After decanting the supernatant, the cell pellet was resuspended in BMMY medium to OD_600_ ≈ 1.0 (in a final volume of 100 mL within a 1 L baffled flask) for protein expression induction. The culture was grown at 28–30 °C with shaking at 300 rpm for 72–120 h with 100% methanol added every 24 h to a final concentration of 0.5% (*v*/*v*) to sustain induction. Supernatants were harvested post-methanol induction by centrifugation (10,000× *g*, 5 min, 4 °C) and filtered through 0.45 μm membranes. The clarified supernatant was loaded onto a 5 mL HisTra™HP column (Smart-Lifesciences, Changzhou, China) pre-equilibrated with binding buffer (50 mM Tris, 500 mM NaCl, pH 7.0). After washing with 10 column volumes (CV) of binding buffer, target xylanase was eluted with 5 CV of elution buffer (250 mM imidazole in binding buffer). The HisTrap flow-through was dialyzed in the buffer (50 mM Tris, 150 mM NaCl, pH 7.0). Protein content was determined spectrophotometrically at 595 nm using the Modified Bradford Protein Assay kit (Sango Biotech, Shanghai, China), with bovine serum albumin as standard.

### 4.3. In-Gel Xylanase Activity Staining

Protein solutions were mixed with SDS loading buffer containing β-mercaptoethanol and incubated at 80 °C for 5 min (non-boiled). Electrophoresis was conducted using 12% SDS-PAGE gels containing 0.2% (*w*/*v*) beechwood xylan (Shanghai Yuanye Bio-Technology Co., Ltd., Shanghai, China) incorporated as follows: A 1% (*w*/*v*) alkaline xylan stock solution was prepared in 0.1 M NaOH and added to the separating gel mixture at 1:5 (*v*/*v*) ratio. The mixture was neutralized with 6 M HCl prior to the addition of ammonium persulfate and TEMED, resulting in a homogeneous suspension of insoluble xylan throughout the gel matrix. Electrophoresis was performed at 4 °C. Then, the gel was subjected to activity staining through the following procedure in a 15 cm glass *Petri* dish: equilibration in 2.5% (*v*/*v*) Triton X-100 for 30 min; two 30 min washes with 50 mM phosphate buffer (pH 5.0); enzyme reactivation in fresh phosphate buffer at 50 °C for 15 min; staining with 0.1% (*w*/*v*) Congo red for 15 min; and destaining with 1 M NaCl for 30–60 min until clear hydrolytic zones appeared against a red background.

### 4.4. Activity Assay of Endoglucanases and Endoxylanases

Endoglucanase and endoxylanase activities were determined using a modified protocol based on [62]. Briefly, 1.5 mL of 1% (*w*/*v*) sodium carboxymethyl cellulose (CMC-Na) or 1% (*w*/*v*) beechwood xylan solution was added to a 25 mL colorimetric tube, followed by the addition of 500 μL of appropriately diluted enzyme solution. The control group uses a heat-inactivated (100 °C, 10 min) enzyme solution. Enzyme dilution should be adjusted to ensure final OD_540_ values (sample OD minus control OD) fell within the optimal range of 0.2–0.8. The enzyme-substrate mixture was incubated for 30 min at their optimal temperatures and pHs. Reactions were terminated by immediate ice-cooling and addition of 3 mL DNS reagent (composition per liter: 10 g 3,5-dinitrosalicylic acid, 20 g NaOH, 200 g sodium potassium tartrate, 2.0 g redistilled phenol, 0.50 g anhydrous sodium sulfite), followed by 10 min boiling. After cooling in an ice-water bath, volumes were adjusted to 25 mL with deionized water (ddH_2_O). Absorbance was measured at 540 nm after thorough mixing to quantify reducing sugars. One unit of enzyme activity was defined as the amount required to liberate 1 μmol glucose or xylose per minute under assay conditions.

### 4.5. pH Profiles of Endoglucanases and Endoxylanases

Optimal pH values were determined by incubating enzyme-substrate mixtures for 30 min at 50 °C in the 0.2 M buffer systems: glycine-HCl (pH 2.0), sodium phosphate-citrate (pH 3.0–8.0), and glycine-NaOH (pH 9.0). pH stability was assessed by pre-incubating purified enzymes in these buffers at 50 °C for 2 h without substrate, followed by residual activity measurement as described above. Untreated enzyme samples served as controls (100% activity reference). All determinations were performed in triplicate.

### 4.6. Temperature Profiles of Endoglucanases and Endoxylanases

Temperature optima were determined at the respective optimal pH values by incubating reaction mixtures for 30 min across a temperature range of 40–80 °C. Thermostability was assessed through 2 h pre-incubation of enzyme solutions at 37–80 °C. Aliquots withdrawn at timed intervals were immediately assayed for residual activity under optimal temperature and pH conditions. Transient thermotolerance of xylanase was evaluated by pre-incubating enzyme solutions at 75 °C, 80 °C, 85 °C, 90 °C, and 95 °C for 1 min. Untreated enzyme samples served as controls (100% activity reference). All assays were performed in triplicate.

### 4.7. Trypsin Resistance Assays

Trypsin resistance was assessed by incubating the TtXyn10C protein in trypsin (from bovine pancreas, DPCC treated) (Sigma-Aldrich, Saint Louis, MO, USA) in 0.2 M Tris-HCl (pH 7.0) at 37 °C with various trypsin/TtXyn10C ratios ranging from 0 to 2000:1 U/mg for 2 h. One unit (U) of trypsin was defined as a 0.001 change in absorbance at 253 nm/min at 37 °C and pH 7.0 with Nα-benzoyl-L-arginine ethyl ester (BAEE) as the substrate. Untreated enzyme samples served as controls (100% activity reference). All assays were performed in triplicate.

## Figures and Tables

**Figure 1 ijms-26-06849-f001:**
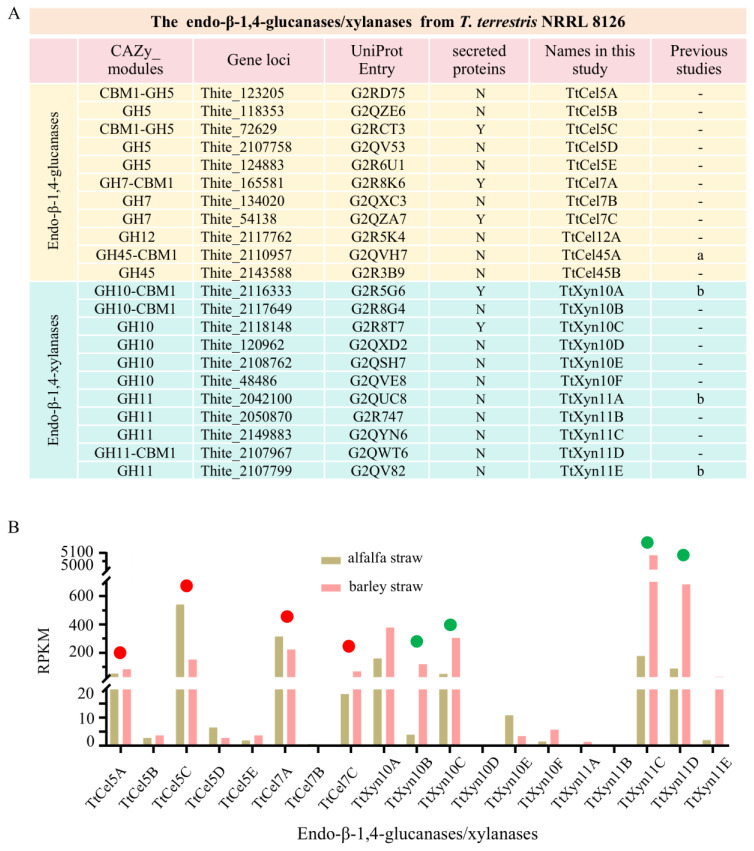
Descriptions of endo-β-1,4-endoglucanases/xylanases from *T. terrestris*. (**A**) Information on CAZyme classification, gene loci, UniProt accession, predicted molecular masses, and nomenclatures in this study. Yellow background, endoglucanases. Blue background, endoxylanases. Y, secreted protein detected; N, no secreted protein detected; –, not reported. a, reference [30]; b, reference [16]. (**B**) The transcription levels of endoglucanases/xylanases genes analyzed by transcriptome data when *T. terrestris* was cultured using alfalfa straw and barley straw as carbon sources. RPKM, reads per kilobase of transcript per million mapped reads. The transcriptome data were retrieved from the Gene Expression Omnibus (GEO) DataSets of NCBI under the accession number GSE27323 [16]. Red and green dots indicate the selected target endoglucanases and xylanases, respectively.

**Figure 2 ijms-26-06849-f002:**
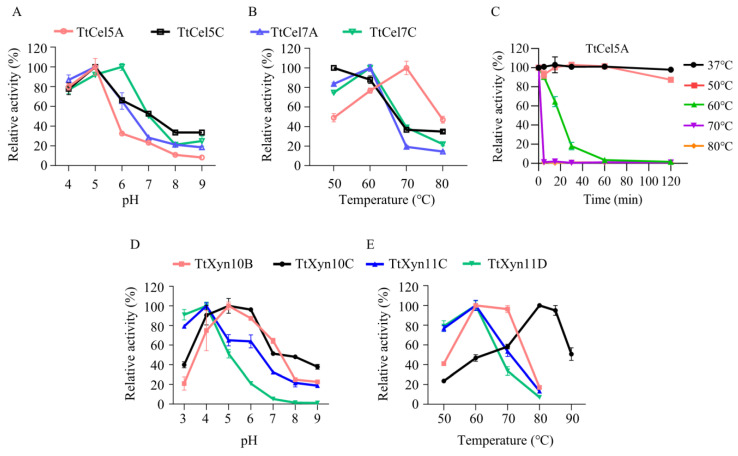
pH and temperature profiling of recombinant endo-β-1,4-glucanases/xylanases. (**A**,**B**) The optimal pH and optimal temperature of endo-β-1,4-glucanases. (**C**) The thermostability of recombinant TtCel5A. (**D**,**E**) The optimal pH and optimal temperature of endo-β-1,4-xylanases.

**Figure 3 ijms-26-06849-f003:**
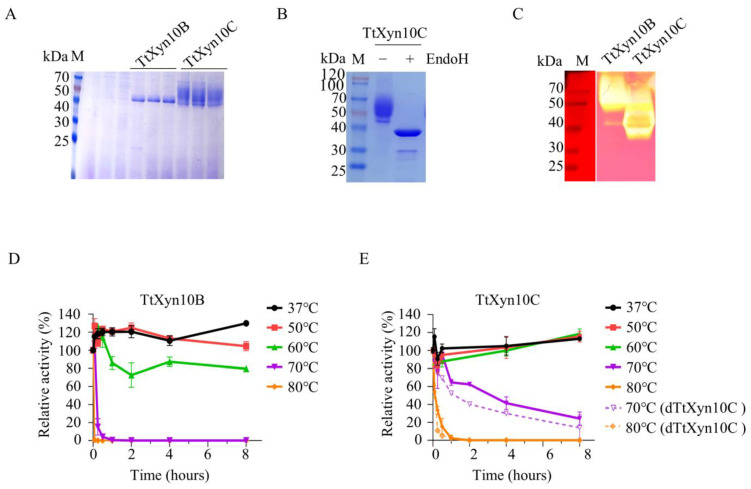
The characterization of recombinant TtXyn10B and TtXyn10C. (**A**) SDS-PAGE analysis of purified TtXyn10B and TtXyn10C. (**B**) TtXyn10C treated by endoH. (**C**) In-gel activity staining using Congo red. Left, adjusting exposure to show the molecular weight marker; right, normal exposure. M, protein molecular weight markers. The thermostability of recombinant TtXyn10B (**D**) and TtXyn10C (**E**), respectively.

**Figure 4 ijms-26-06849-f004:**
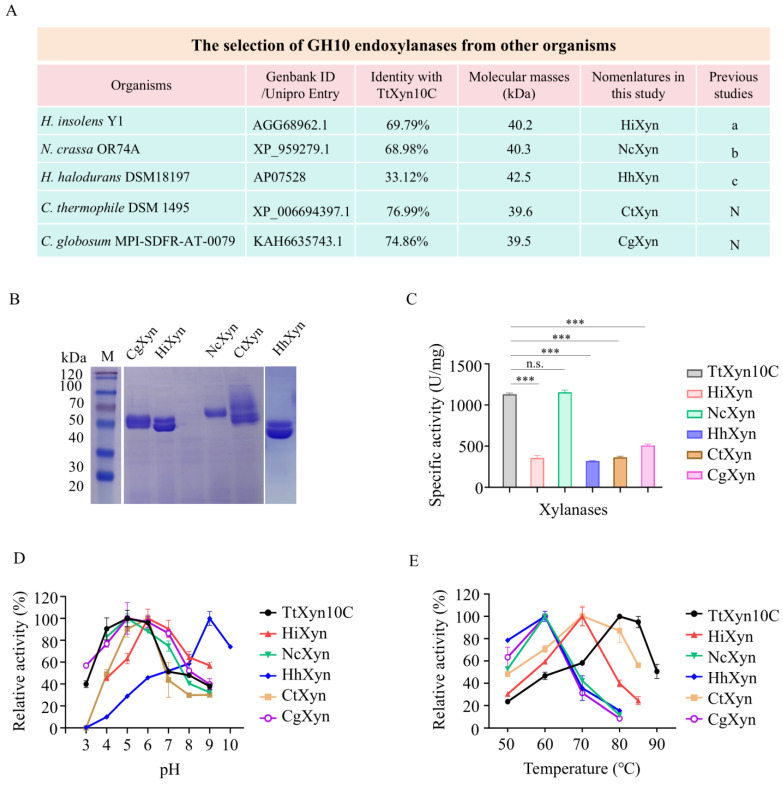
The comparison of TrXyn10C with other GH10 xylanases. (**A**) The selection of GH10 xylanases from other species. a, reference [31]; b, reference [33]; c, reference [32]. N, not reported. The SDS-PAGE (**B**), specific activities (**C**), optimal pHs (**D**), and optimal temperatures (**E**) of purified recombinant xylanases. M, protein molecular weight markers. n.s., not significant. *** *p* < 0.001.

**Figure 5 ijms-26-06849-f005:**
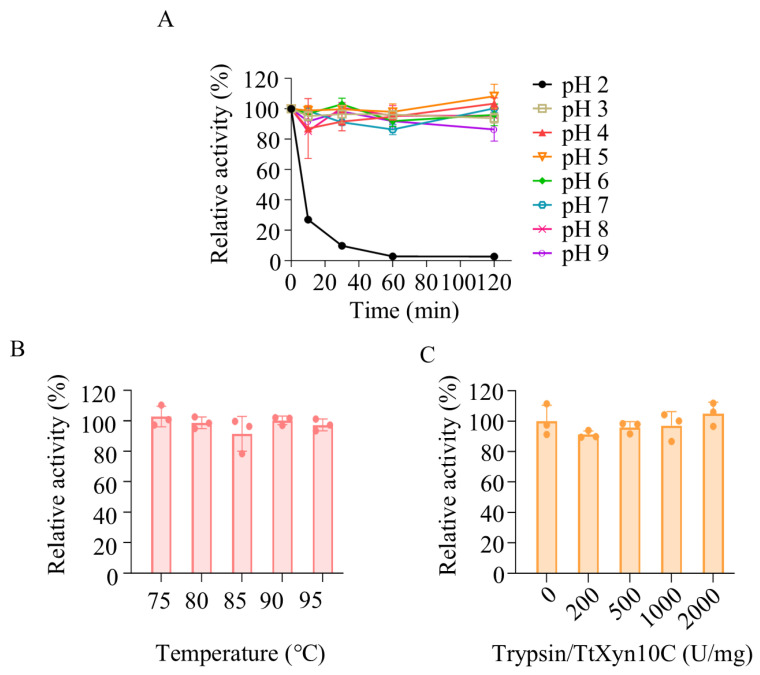
Determination of pH stability (**A**), transient high-temperature stability (**B**), and trypsin stability (**C**) of TrXyn10C. Trypsin was added at various trypsin (U)/TrXyn10C (mg) ratios ranging from 200:1 to 2000:1.

**Figure 6 ijms-26-06849-f006:**
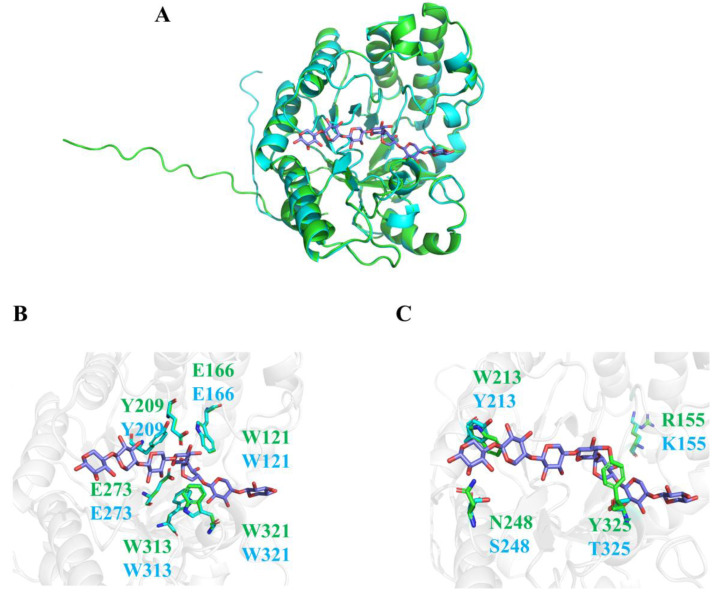
Overall structures and active-site architectures of TtXyn10C and CtXyn. (**A**) Structural models of TtXyn10C (blue) and CtXyn (green) complexed with xyloseptaose substrate. Molecular graphics depicted using PyMOL. (**B**) Conserved residues around subsites −1 and +1. (**C**) Divergent residues within 5 Å of the substrate in the active site. TtXyn10C residues labeled in blue; CtXyn residues labeled in green.

## Data Availability

Data are contained within the article.

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
