# Peer review of "Screening, Characterization and Comparison of Endoglucanases/Xylanases from Thermophilic Fungi: A Thielavia terrestris Xylanase with High Activity-Stability Properties"

_ijms, 2025, doi:10.3390/ijms26146849_

Round 1
Reviewer 1 Report
Comments and Suggestions for Authors
The article "Screening, characterization, and comparison of endoglucanases/xylanases from thermophilic fungi: a Thielavia terrestris xylanase with high activity-stability properties" is aimed at studying the characteristics of various endo-β-1,4-glucanases/xylanases. The authors studied in detail the optimal temperature and pH of various endo-β-1,4-glucanases/xylanases, their thermal stability and resistance to trypsin. The article is suitable for the publication of the International Journal of Molecular Sciences, but there are several comments that should be addressed before publication.
1. In the abstract, I would like to see what is the novelty of this study?
2. In the introduction, out of 25 references, 10 are from the last 5 years, and only 1 reference is from 2025. It is necessary to show the relevance of this work by adding references for 2024-2025.
3. I would like to see in the discussion of the results in what areas of industry the authors plan to use the obtained enzymes with high stability? And how will these improved properties positively affect the corresponding process?
Author Response
Thank you for your valuable comments on our manuscript.
We have carefully read your comments and revised the manuscript accordingly. Our detailed responses can be found in the attached PDF file.

Reviewer 2 Report
Comments and Suggestions for Authors
This study comprehensively characterizes lignocellulose-degrading enzymes from the thermophilic fungus Thielavia terrestris, focusing on endoglucanases (GH5, GH7) and endoxylanases (GH10, GH11). Through transcriptomics and secretome analysis, eight enzymes (four endoglucanases, four xylanases) were selected for heterologous expression in Pichia pastoris. Below are my comments:
- TtXyn10C shows abnormal band spreading (45-70 kDa) versus predicted 39.6 kDa. While glycosylation is proposed (Fig 3B), direct evidence is lacking. Perform deglycosylation assays and quantify modification extent.
- Specify if DNS method controls (e.g., no-enzyme blanks) were used to exclude background reducing sugars; Confirm whether substrate saturation was achieved for fair activity comparisons.
- The claim that "20% of T. terrestris enzymes are thermostable" relies on limited data (3/12 enzymes). Reframe as preliminary observation.
- Contrast TtXyn10C’s sequence with unstable homolog CtXyn10C (76.99% identity) to highlight potential stability determinants.
- Update enzyme stability-activity trade-off discussion with recent GH engineering studies (e.g., 2023-2025).
- What specific structural adaptations allow TtXyn10C to maintain both high activity and stability when its close relative CtXyn10C fails under heat?
- Could TtXyn10C work with other thermostable enzymes from terrestris(e.g., GH45 cellulase) to break down complex biomass?
Author Response

(The authors gave the same response as above.)

Reviewer 3 Report
Comments and Suggestions for Authors
The article is a comprehensive study of thermostable endoglucanases and xylanases isolated from thermophilic fungus Thielavia terrestris. The main focus is on the characterization of TtXyn10C xylanase, which demonstrates exceptional thermal stability and high catalytic activity. The work is well structured, the methods are described in detail, and the results are confirmed by experimental data. The work compares TtXyn10C with other known xylanases from various organisms, which allows us to evaluate its advantages. For example, TtXyn10C surpassed the activity and thermal stability of enzymes from Humicola insolens and Halalkalibacterium halodurans. The authors propose the use of TtXyn10C in the feed industry, where resistance to high temperatures (for example, during feed granulation) and a wide pH range are important. This makes the research not only theoretically significant, but also practically useful. However, there are several shortcomings in the article that require attention.
1 The authors write that TtXyn10C undergoes glycosylation, but does this affect its activity and stability?
2 Can you provide kinetic parameters (e.g. Km and Vmax) for the studied enzymes, as this is important for understanding their catalytic efficiency
3 When determining enzyme activity, no standard deviations were specified or no statistical analysis of the significance of the differences was performed.
4 In the Methods it is necessary to describe in detail the conditions of cultivation of the studied strain.
5 Is it possible to analyze hydrolysis products to understand the mechanism of action of enzymes using HPLC?
6. Was protease contamination monitored when determining enzyme activity?
7. Has protein aggregation been analyzed at high temperatures?
8. Are you going to conduct a phylogenetic analysis of the domains or sequences of amino acids responsible for thermal stability?
9.Can you make a guess why TtXyn10C has more activity?
10. Figure 4E - HiXyn line - why is there such a huge error at 70 degrees?
11. Fig. 5A - it is necessary to change the scale on the graph, the pH 3-9 lines merge into one.
12. It is necessary to make all images more visual. Everything is too small.
Author Response

(The authors gave the same response as above.)

Round 2
Reviewer 3 Report
Comments and Suggestions for Authors
The authors substantially revised the article and took into account the reviewer's comments. The article can be published in its current form.